# A Mathematical Model for Determining the Body’s Fluctuating Need for and Synthesis of Active Vitamin D

**DOI:** 10.3390/biomedicines11020324

**Published:** 2023-01-24

**Authors:** Sean R. Maloney

**Affiliations:** Department of Rehabilitation Medicine, W.G. (Bill) Hefner VA Medical Center, 1601 Brenner Avenue, Salisbury, NC 28144, USA; sean.maloney@va.gov; Tel.: +1-336-406-6626 or +1-704-638-3468; Fax: +1-704-638-3811

**Keywords:** 25-hydroxyvitamin D, 1,25-dihydroxyvitamin D, immune system receptors, inflammatory response, COVID-19

## Abstract

The process by which 1,25(OH)_2_D_3_ is synthesized and degraded and how it is transported out of the cell and body is described. The changing demand for the synthesis of 1-25(OH)_2_D_3_ during different conditions experienced by the body is reviewed. A method of determining 1,25(OH)_2_D_3_ synthesis and demand, and the percent utilization of 25(OH)D_3_ to make 1,25(OH)_2_D_3_ is presented based on the measurement of the end metabolites of 1,25(OH)_2_D_3_ and of its immediate precursor, 25(OH)D_3_. A mathematical model has been developed to allow the calculation of 1,25(OH)_2_ D synthesis, and demand, and the percent utilization of 25(OH)D_3_. Simple algebraic equations have been derived which allow the calculation of these new parameters using the concentrations of the end metabolites of 1,25(OH)_2_D_3_ and its immediate precursor, 25(OH)D_3_ in the serum and urine. Vitamin D plays an important role in combating invading bacteria and viruses and in subduing the body’s associated inflammatory response. This new approach to evaluating vitamin D status may help clinicians determine 25(OH)D_3_ and 1,25(OH)_2_D_3_ levels needed to suppress bacterial infections, viral replication during new viral infections and the reactivation of latent viruses, and to downregulate the inflammatory responses caused by bacteria and viruses.

## 1. Introduction

Vitamin D is a fat-soluble vitamin and a hormone that has receptors in many tissues of the body. It plays an important role in many different normal and abnormal conditions that routinely affect the human body. One of the first roles that was discovered for vitamin D was the maintenance of normal calcium and phosphorous levels in bone in order to prevent rickets in children and osteoporosis in adults. It is necessary for maintaining normal muscle strength. It has a very important role in immune system function in order to fight off infection and modulate the inflammatory response to infection. It is important for a developing embryo and fetus during pregnancy. Finally, there is growing evidence that it is important to reduce the risk of heart disease, type 2 diabetes, cancer, and the risk of premature death and cognitive decline [1].

There have been numerus studies recently which have linked increased COVID-19 morbidity and mortality to low vitamin D states [2,3,4,5,6]. Likewise, there have been recent review articles which discuss reasons why low vitamin D states may play a role in a patient’s ability to fight off or mitigate the morbidity and mortality associated with a COVID-19 infection [7,8,9]. Central to this new focus on vitamin D status is the important role that vitamin D plays in the modulation of the immune system [10,11,12]. Immune system cells with vitamin D receptors exist in large numbers in many tissues of the body including primary lymphoid organs (bone marrow and thymus) and secondary lymphoid organs (lymph nodes, the spleen, the tonsils, skin, and various mucous membrane layers in the body including those of the nose, throat, and bowel) [13]. An important recent article reviews the relationship between co-morbidities in COVID-19 patients known to be associated with increased morbidity and mortality and those same co-morbidities in low vitamin D states [14].

Persistent low vitamin D states may chronically impair the immune system in individuals in two ways. First, in low vitamin D states, the immune system may be unable to maximally suppress viruses such as COVID-19 in part due to inadequate production of cathelicidin and defensin β2 [15,16,17]. Second, vitamin D is a hormone which acts directly on the immune system including B and T lymphocyte cells to downregulate the inflammatory reaction triggered by viral antigens, or other microorganisms [18].

Plasma 25(OH)D_3_ vitamin D precursor levels represent vitamin D stores in the body and have been traditionally measured to assess vitamin D status. This paper will build on and review evidence supporting the following two premises: (1) plasma 25(OH)D_3_ levels do not consistently represent 1,25(OH)_2_D_3_ synthesis (or total body levels), demand for 1,25(OH)_2_D_3_, or percent utilization of 25(OH)D_3_ to form 1,25(OH)_2_D_3_ by the body, and (2) the need for and utilization of active vitamin D by tissues of the body can vary between individuals and dramatically in the same individual under different circumstances. Many factors can reduce a person’s ability to produce active vitamin D including low vitamin D precursor stores, increased body mass index, BMI (i.e., obesity) which dilutes the concentration of vitamin D precursor in the serum, decreased exposure to ultra-violate B, UVB, light, increasing age, exogenous medications, and genetic factors.

The measurement of active vitamin D, 1,25(OH)_2_D_3_, plasma levels is possible and in certain situations is indicated. This paper will present information to support the premise that serum active vitamin D levels often do not correlate with vitamin D precursor levels and may not correlate well with active vitamin D synthesis in the body.

## 2. Methods

### 2.1. Vitamin D Molar Balance Model

A molar balance approach to analyze active vitamin D synthesis and degradation is presented in this paper. Instead of focusing on the front end (input) of the vitamin D synthesis pathway, this paper will focus only on the final metabolism of active vitamin D and its immediate precursor. This molar balance approach may permit (1) an estimation, at one point in time (t), of the amount of active vitamin D that has been recently produced in the body, (2) the determination (based on measurable quantities) of the synthesis of active vitamin D, 1,25(OH)_2_D_3_, over 24 h, and (3) the evaluation of the body’s overall demand for 1,25(OH)_2_D_3_ at a given point in time (t). Overall demand will be determined by comparing the amount of active vitamin D precursor, 25(OH)D_3_, used to synthesize active vitamin D to the amount of active vitamin D precursor, that is diverted away from the synthesis of active vitamin D and metabolized (wasted). Knowing the demand for 1,25(OH)_2_D_3_, and the percent utilization of active vitamin D precursor to produce active vitamin D, may help to determine whether inadequate, adequate, or excess vitamin D precursor is present. Measurement of the above vitamin D characteristics may enable clinicians to better identify individuals who may have impaired immune system function associated with inadequate active vitamin D synthesis and to better treat these individuals when they have COVID-19 viral or other acute infection infections. Similarly, measurement of these characteristics may allow clinicians to better treat those individuals with chronic inflammation associated with partially reactivated, latent viruses.

The 25(OH)D_3_ molar balance model is depicted qualitatively in Figure 1. This figure provides a diagrammatic picture of the movement of vitamin D through the body and between the intracellular and extracellular compartments of the body. The blue arrow in Figure 1 depicts an intracellular synthesis pathway for 1,25(OH)_2_D_3_ without its immediate precursor, 25(OH)D_3_, ever having to pass through or to be initially stored in the extracellular plasma space.

The molar balance model is described quantitatively in several equations included in this text as well as in a more rigorous manner in the Appendix A. The figures and equations included in this text along with the Appendix A describe the active vitamin D precursor (25(OH)D_3_) transport, conversion to active vitamin D, metabolism, and excretion from the body. In this molar balance model, the body is divided into two main compartments for simplicity. The extracellular compartment is made up of all those spaces in the body which are not made up of cells and which contain fluids such as plasma, lymphatic fluid, bile, interstitial fluid, etc. The intracellular compartment is the space made up all those cells in the tissues of the body. It is important to remember that the synthesis of active vitamin D and its breakdown to its end metabolites is an intracellular process. See Figure 2a,b below.

For the purpose of this model, the spaces which are made up of urine in the bladder and stool in the distal small bowel and large bowel are considered outside the body and the end metabolites of vitamin D are assumed lost to the intracellular and extracellular spaces of the body once they enter urine and stool in these spaces. The cells of bladder and bowel tissues including the cells of their mucosal linings are in the intracellular space/compartment of the body (see Figure 2c).

This model describes the synthesis and transport of active vitamin D, 1,25(OH)_2_D_3_, and its inactive metabolites into and out of the intracellular and extracellular compartments of the body at any arbitrary time (t) and over an arbitrary, 24 h, interval of time. The relationship between the end inactive metabolites of 1,25(OH)_2_D_3_ and the end inactive metabolites of 25(OH)D_3_ (that are diverted away from 1,25(OH)_2_D_3_ synthesis, i.e., wasted) are used to establish changing 1,25(OH)_2_D_3_ demand and changing vitamin D precursor, 25(OH)D_3_, utilization by the body to synthesize active vitamin D.

This model may also provide a better understanding of why: (1) the same store or concentration of 25(OH)D_3_ in the plasma may result in different overall (increased or decreased) rates of 1,25(OH)_2_D_3_ synthesis in cells of the tissues of the body, (2) the same store or concentration of 25(OH)D_3_ in the plasma may result in an inadequate rate of 1,25(OH)_2_D_3_ synthesis for one individual but not for another or for one individual experiencing increased 1,25(OH)_2_D_3_ demand over their usual vitamin D demand state [19,20,21]. and (3) a significant portion of D_3_ can be converted from vitamin D_3_ to 25(OH)D_3_ and then to 1,25(OH)_2_D_3_ in the same cell without this portion of the body’s vitamin D precursor, 25(OH)D_3_, ever passing through the plasma (sub-extracellular) space before being converted to 1,25(OH)_2_D_3_ [22]. The active form of vitamin D is a hormone and is structurally different from its precursor molecules and its inactive metabolites. This difference allows active vitamin D to attach to receptor sites in or on target cells in order to activate different genes and chemical reactions. As a result of a body’s changing demand or need for active vitamin D, cells/tissues in the body can produce increasing or decreasing amounts of the active form of vitamin D, 1,25(OH)_2_D_3_, from its precursor, 25(OH)D_3_. The source of vitamin D precursor needed for this synthesis can come from vitamin D precursor already present in the body’s cells/tissues, or from vitamin D precursor entering the body’s cells/tissues from the plasma space.

The changing need for active vitamin D can occur in cells/tissues which cannot completely synthesize their own active form of vitamin D but rely on plasma vitamin D precursor, 25(OH)D_3_, or rely on active vitamin D, 1,25(OH)_2_D_3_ from other cells/tissues of the body (endocrine source). There are also the cells/tissues that can synthesize their own active vitamin D (autocrine or paracrine source for active vitamin D). Thus, some cells/tissues that store vitamin D precursors, can produce their own active vitamin D independently of the extracellular plasma source of the vitamin D precursor, 25(OH)D_3_.

Changing demand and synthesis of active vitamin D has been documented in several studies. One recent study which compared active vitamin D serum levels in non-pregnant women with those in pregnant women found significantly increased active vitamin D levels in those who were pregnant. Mean levels of active vitamin D, 1,25(OH)_2_D_3_, at 15 weeks were as follows: non-pregnant controls, 85.6 picomoles per liter (pmols/L), non-preeclamptic pregnant women, 336.3 pmols/L, and preeclamptic pregnant women 388.8 pmols/L [23]. Mean serum 25(OH)D_3_ levels in the same groups, respectively, were 46.8 nmols/L, 44.7 nmols/L, and 33.1 nmols/L. In the first two groups of women, mean serum 25(OH)D_3_ levels were similar, but mean serum 1.25(OH)_2_D_3_ levels varied by a factor of 4.

In comparing the mean serum 25(OH)D_3_ levels of the last two groups of pregnant women, the pre-eclamptic women had a lower mean serum 25(OH)D_3_ level but a higher mean serum 1,25(OH)_2_D_3_ level. These findings represent a dramatic example of uncoupling of mean serum 25(OH)D_3_ and 1,25(OH)_2_D_3_ levels when comparing non-pregnant and pregnant women’s levels and to a lesser extent when comparing pregnant and pre-eclamptic pregnant women’s levels.

An increase in vitamin D utilization/demand has been documented by serial measurements of dropping serum vitamin D precursor, 25(OH)D_3_ levels during military training of short duration [24]. During a 1993 study of nutritional status in young healthy US Army soldiers undergoing an arduous 21-day Special Forces Evaluation and Selection Program, serum vitamin D precursor, 25(OH)D_3_ levels dropped from an initial mean level of 61 ng/mL (range 34–100 ng/mL) on the first day to a mean level of 55 ng/mL (range 38–97 ng/mL) on day 10, to a mean level of 51 ng/mL (range 42–60) by day 20. The statistical significance of this drop was not tested at the time perhaps in part because vitamin D was only one of many nutritional status biomarkers that were measured and because mean vitamin D levels remained above the upper limits of normal (50 ng/mL at the time). To approximately convert ng/mL to nmols/L, multiply ng/mL by 2.5.

Vitamin D binding protein and total and free vitamin D metabolites have been shown to have a changing or diurnal rhythm (i.e., repeated daily pattern of change in vitamin D metabolite levels during the day) [25]. A recent case study of a middle-aged woman, who was taking 5000 IUs of vitamin D_3_ orally each afternoon for over one year, demonstrated not only a daily serum 25(OH)D_3_ diurnal rhythm but also a temporary significant drop in serum 25(OH)D_3_ level associated with an acute but short duration respiratory illness [26]. The study included four separate days of evaluation, 11 October, 18 October, 15 November, and 28 November 2017) and the midday serum 24(OH)D_3_ levels on each of these days was: 67 ng/mL, 65 ng/mL, 51 ng/mL (at the time of a brief cold), and 67 ng/mL, respectively. The approximate drop of 25% on day three was significant, *p* < 0.013.

The level of vitamin D precursor, 25(OH)D_3_, in the plasma is often not proportional to the level of active vitamin D, 1,25(OH)2D_3_ in the plasma and does not directly indicate the rate of active vitamin D synthesis in all the tissues of the body. This occurs in part because paracrine/autocrine 1,25(OH)_2_D_3_ can be synthesized and metabolized in cells/tissues of the body without active vitamin D precursor, 25(OH)D_3_, passing from the plasma into the affected cell and without paracrine/autocrine produced active 1,25(OH)_2_D_3_ passing from the tissue/cells where it is produced and consumed into the plasma space. This observation is supported by the concurrent presence of the enzymes that allow the conversion of D_3_ to 25(OH)D_3_ and the enzyme that allows the conversion of precursor, 25(OH)D_3_, to active vitamin D, 1,25(OH)2D_3_, in some cells/tissues of the body. The enzymes (CYP2R1, CYP3A4, and CYP27A1) which convert D_3_ to 25(OH)D_3_, the enzyme (CYP27B1) which converts 25(OH)D_3_ to 1,25(OH)_2_D_3_, and the enzyme (CYP24A1) which converts 25(OH)D_3_ and 1,25(OH)_2_D_3_ to inactive forms of vitamin D (i.e., 24,25(OH)_2_D_3_, 23,25(OH)_2_, 1,24,25(OH)_3_D_3_ and 1,23,25(OH)_3_D_3_) are described in more detail in other publications [27,28].

The RNA expression of these enzymes has been examined recently in 27 different tissues of 95 human subjects as part of a larger study of gene expression [29]. The units of gene transcript expression are RPKM (i.e., Reads per Kilobase of transcript, per Million mapped reads) [30]. In comparing/interpreting the RPKM reads, (i.e., comparing the ability of cells/tissues to produce 25(OH)D_3_ and its metabolites), it is important to note that RPKM reads represent the density of RNA in the tissue sample and not the overall size of the tissue or organ. Thus, the ability of tissues and organs to produce enzymes or process substrate will also depend on the total relative size of the tissue or organ and other factors such as the rate of diffusion or perfusion of the substrates through the tissue. The following four paragraphs outline the presence of the enzymes of vitamin D metabolism in many of the cells/tissues of the body.

The CYP27A1 enzyme had the greatest representation in the body among the three enzymes which convert D_3_ to 25(OH)D_3_ [31]. Of the 27 tissues studied, liver was the tissue with the highest expression of this gene (approx. 101.4 RPKM), followed by small intestine (27.6 RPKM), lung (approx. 26 RPKM), and duodenum (approx. 23 RPKM). However, most of the other tissues examined had significant expression of this enzyme RNA including prostate (18 RPKM), kidney (18 RPKM), adrenal (12 RPKM), ovary (11 RPKM), brain-colon-urinary bladder (9 RPKM), lymph node tissue (8 RPKM) and fat (6 RPKM). The second most widely distributed enzyme RNA (but of lower concentration) for this first conversion, D_3_ to 25(OH)D_3_, was CYP2R1 [32]. This enzyme was most highly expressed in skin (2.2 RPKM). Other tissue representation included testes (2.0 RPKM), duodenum (approx. 1.27 RPKM), small intestine (1.25 RPKM), appendix (1.36 RPKM), lymph node tissue (1.1 RPKM), spleen (1.0 RPKM), fat and thyroid (0.9 RPKM) and bone marrow (0.45 RPKM). The third enzyme for the conversion of D_3_ to the vitamin D_3_ precursor, 25(OH)D_3_ was CYP3A4 [33]. This enzyme was largely expressed in only three tissues: liver (476.5 RPKM), small intestine (282.8 RPKM), and duodenum (approx. 250 RPKM).

Enzyme CYP27B1 converts 25(OH)D_3_ to the active form of vitamin D, 1,25(OH)_2_D_3_ [34]. Its RNA was present in almost all the tissues studied but most significantly present in kidney (9.5 RPKM) and thyroid (4.8 RPKM), followed by appendix (approx. 0.9 RPKM), lymph node (0.8 RPKM), bone marrow and adrenal tissue (0.4 RPKM), and fat (0.1 RPKM).

CYP24A1 is the enzyme which metabolizes both 25(OH)D_3_ and 1,25(OH)_2_D_3_ to the inactive forms 24,25(OH)_2_D_3_, 23,25(OH)_2_D_3_, 1,24,25(OH)_3_D_3_, and 1,23,25(OH)_3_, respectively, as well as to their end metabolites [28,29,30,31,32,33,34,35]. The catabolic CYP24A1 enzyme RNA is present in many tissues, and is highly present in urinary bladder (approx. 21.5 RPKM), and endometrium (10.7 RKPM), followed by kidney (approx. 3.5 RPKM), and placenta (approx. 2.5 RPKM). The high levels of CYP24A1 RNA and potential for this enzyme’s synthesis in the endometrium and placenta may be required due to the increased 1,25(OH)_2_D_3_ synthesis during pregnancy (see Figure 2d).

The RNA expression study from which the above data are taken did not include skeletal muscle tissue, and it is not known if enzyme RNA, for both steps in the conversion of D_3_ to 1,25(OH)_2_ D_3_, exists in skeletal muscle. However, CYP27B1 has been found to be present in skeletal muscle [36,37,38]. In this molar balance analysis, D2 and its metabolites, and C-3-epi-25(OH)D_3_ and 1,25(OH)_2_-3-Epi-D_3_ are ignored to simplify the model [39].

The concurrent existence of DNA genes and their resulting RNA that code for enzymes capable of converting D_3_ to 25(OH)D_3_ (3 possible enzymes) and 25(OH)D_3_ to 1,25(OH)_2_D_3_ (1 enzyme) in many different tissue locations suggests the existence of a synthesis pathway from D_3_ to 1,25(OH)_2_D_3_ in cell/tissue types that are independent of the need for transported extracellular (plasma) compartment 25(OH)D_3_ generated from the liver. The presence of RNA for a particular enzyme does not necessarily equate to a functional protein being translated from the RNA and the functionality of the proteins would need to be verified. The large presence of immune system cells in many tissues of the body makes the immune system potentially one of the largest producers and consumers of active vitamin D during certain conditions such as severe infections especially involving multiple tissues such as with COVID-19 virus infections.

An alternate, strictly intracellular, 1,25(OH)_2_D_3_ synthesis and degradation pathway supports the idea that extracellular (plasma) compartment 25(OH)D_3_ and 1,25(OH)_2_D_3_ levels are partially uncoupled from intracellular compartment active vitamin D synthesis. An intracellular active vitamin D synthesis pathway independent of plasma, 25(OH)D_3_, may explain why the rate of active vitamin D, 1,25(OH)_2_D_3_, synthesis is partially independent from the aqueous (plasma) vitamin D precursor, 25(OH)D_3_ concentration.

Serum 25(OH)D_3_ levels and 1,25(OH)_2_D_3_ levels do not appear to have a proportional relationship, and serum 25(OH)D_3_ concentration in the plasma space may not necessarily reflect the body’s ability to meet the need for active vitamin D synthesis. The following mathematical equations for the molar balance analysis are based on and depend on the accurate measurement of changing concentrations of vitamin D end metabolites in plasma and urine samples. This model can be used to estimate active vitamin D synthesis and to assess whether the body’s demand for vitamin D is being met. This molar balance method, of assessing vitamin D status, analyzes and depends on primarily the backend of the vitamin D metabolic pathway rather than the frontend.

Some data concerning where vitamin D_3_ and 25(OH)D_3_ are stored in the body exist based on intravenous radiolabeled vitamin D studies. D_3_ has been shown to accumulate the most in adipose tissue but D_3_ also accumulates in muscle, skin, plasma, and other organs. Total body storage of vitamin precursors has been estimated to be approximately 1 micro-mole (10^−6^ moles) with two-thirds stored as D_3_ (primarily in adipose tissue) and the rest primarily stored as 25(OH)D_3_ in various tissues and fluids including adipose tissue 35%, muscle 20%, and plasma/extracellular fluid 30% [40,41]. The molar balance model divides the storage of vitamin D into two compartments: intracellular and extracellular. Plasma is considered a sub-space of the extracellular compartment. The concentrations of 25(OH)D_3_, 1,25(OH)_2_D_3_ and their end metabolites are assumed to be approximately uniform or homogeneous in both the plasma and total extracellular spaces. In contrast, levels of vitamin D precursors and active vitamin D are not assumed to be homogeneous within the cells/tissues of the intracellular compartment or spaces.

Using radioactive H^3^ tagging, attempts have been made in several studies to determine the portion of vitamin D and its metabolites excreted in the urine verses in the stool (through bile) [42,43,44]. In three studies approximate differences between excretion in the urine verses in the stool have been as follows: 32% versus 68% (n = 7), 25% versus 75% (n = 7), and 22% versus 78% (n = 16). The molar balance model uses the values of 25% excretion in the urine and 75% in the stool.

### 2.2. Model Assumptions

1.Constant Compartment Volumes

The volume of the body’s extracellular compartment or space, V_e_, (or plasma sub-compartment, V_p_) can change in certain circumstances such as pregnancy, severe dehydration, or congestive heart failure. However, for our model, it is assumed that the extracellular (or plasma) volumes at some initial time, t_0_, V_e_(t_0_), equals the extracellular compartment volume at some later time, t_1_, V_e_(t_1_), and equals a constant V_e_. Similarly, the body’s intracellular volume, V_i_ is assumed to be constant [45,46]

2.Intracellular volume = 2 × Extracellular Volume

For a complete list of different volume assumptions for men and women, see Table 1, “Intracellular and Extracellular Compartment Volume Analysis”, and Table 2, “Total Male and Female Intracellular, Extracellular, and Plasma Volumes”.

3.The C_n_ Intracellular Concentration of End Metabolites = β Times the C_n_ Extracellular Concentration

The mean intracellular concentrations of the end metabolites, m_n_, will be assumed to be proportional to and a factor β times the same concentrations in the extracellular compartment. The β factor is greater than 1 in this model because the end metabolites are assumed to be transported from the intracellular compartment (where they are made) to the extracellular compartment by a concentration gradient in order to be excreted from the body (see Figure 2d).

4.Extracellular Sub-Compartment C_n_ Concentrations Are Equal

Vitamin D metabolite, m_n_, concentrations may differ somewhat between extracellular sub-compartments/spaces and their concentration gradients around cells may vary. However, in our model, the concentrations of 25(OH)D_3_ and its many metabolites (including 1,25(OH)_2_ D_3_) are assumed to be uniform for the same metabolites across different extracellular sub-compartments at any given point in time.

5.Nanomoles, 10^−9^ Moles (nmols) of Active Vitamin D Synthesized over 24 h Equals the nmols of Active Vitamin D End Metabolites Synthesized over the Same 24 h

The Molar Balance Model for active vitamin D synthesis assumes that the number of nmols of active vitamin D synthesized over a 24 h period equals the sum of the number of nmols of end metabolites of active vitamin D synthesized over the same 24 h. This steady state assumption further depends on the following: (1) any change in the total body’s number of nmols of active vitamin D or in the number of nmols of active vitamin D intermediate metabolites in the intracellular plus extracellular compartments is small compared to the synthesis of active vitamin D end metabolites over 24 h, and (2) the number of nmols of active vitamin D and active vitamin D intermediate metabolites excreted is relatively small compared to the nmols of end metabolites excreted over the same 24 h period. All the measurements made using this and the following assumptions are in units of nmols. The fact that this measurement is made over 24 h allows this measurement also to represent active vitamin D synthesis per 24 h (a synthesis rate).

The equations derived to determine the nmols of active vitamin D synthesized over 24 h depend on the above assumptions. The validity of these assumptions could be checked by doing second serum and urine assays which measures total end metabolites of active vitamin D precursor, active vitamin D, and all their intermediate metabolites after all active vitamin D precursor, active vitamin D, and its intermediate metabolites are driven to their end metabolites by adding significant excess CYP24A1 hydroxylase enzyme. If, after adding measured nmols of serum active vitamin D and its precursor, 25(OH)D_3_, to the measurement of their end metabolites in the first sample, a discrepancy between the sum of these metabolites in the first sample and the total end metabolites of 25(OH)D_3_ and 1,25(OH)_2_D_3_ in the second sample exists, then a correction factor can be added to the calculation (see further discussions below and in the Appendix A, Section 2, Assumptions).

6.The nmols of End Metabolites, m_8:10_, in the Intracellular plus Extracellular Spaces Can Be Used to Estimate the nmols of Active Vitamin D Synthesis (Levels) at Time (t).

The total nmols of active vitamin D, 1,25(OH)_2_D_3_ end metabolites (m_8_, m_9_, and m_10_) in the extracellular plus intracellular spaces, can be used to estimate the body’s total active vitamin D synthesis at any point in time, (t). The assumption in the sentence above allows for a quicker method for assessing active vitamin D synthesis at any given time (t) from a single plasma sample and to track changes in active vitamin D synthesis due to exogenous vitamin D supplementation. The use of only the end metabolites of active vitamin D as a biomarker for active vitamin D synthesis only approximates active vitamin D synthesis when the levels (synthesis) of active vitamin D and its intermediate metabolites are changing significantly (going up or down) as the demand and synthesis of active vitamin D changes.

As with the 24 h assessment above, a second plasma serum assay could be performed which measures the total end metabolites of active vitamin D after all active vitamin D and its intermediate metabolites are driven to their end metabolites by the CYP24A1 hydroxylase enzyme. This assay could be used to check for a significant buildup of active vitamin D intermediate metabolites relative to active vitamin D end metabolites. If the discrepancy is large then a factor to correct for this discrepancy must be added to the molar balance model calculation. This same analysis could be performed on a spot urine although the dilution variability of spot urines would have to be accounted for (see Figure 2a,b and Assumptions (A.8) and (A.9) and their description in Appendix A).

7.The Total End Metabolites of 25(OH)D_3_ in the Extracellular and Intracellular Spaces That Are Not Metabolized from 1,25(OH)_2_D_3_ Represent the nmols of Vitamin D precursor, 25(OH), That Are Diverted away from the Synthesis of Active Vitamin D at Any Point in Time (t)

Determination of the total nmols of m_6_ and m_7_ (end metabolites of m_0_) in the extracellular and intracellular spaces at time (t) are assumed to represent the total number of nmols of 25(OH)D_3_ diverted away from the synthesis of active vitamin D at time (t). This assumption further assumes that the nmols of diverted 25(OH)D_3_ (m_0_) intermediate metabolites are small in number compared to the end metabolites of diverted 25(OH)D_3_ (m_0_). The assumption in the sentence above allows for a quicker method (1) to assess the total nmols of vitamin D precursor, 25(OH)D_3_ diverted away from the synthesis of active vitamin D at any given time (t) from a single plasma sample and (2) to track changes in active vitamin D precursor diversion following exogenous vitamin D supplementation.

The validity of this assumption can be checked by doing a second serum assay from a plasma sample drawn at time (t) which measures total end metabolites of active vitamin D precursor, m_0_, after 25(OH) vitamin D and all active vitamin D precursor intermediate metabolites are driven to their end metabolites by CYP24A1 hydroxylase enzyme. In this case, the number of nmols of active vitamin D precursor, m_0_, originally in the serum level of the plasma sample must be subtracted from the measured diverted end metabolites of m_0_. The diverted nmols of active vitamin D are used in the calculation of the active vitamin D demand ratio and in the determination of the percent utilization of active vitamin D precursor, m_0_, in the synthesis of active vitamin D.

8.This Model Assumes That 25% of 1,25(OH)_2_D_3_ and Its Metabolites Are Excreted in the Urine and 75% Are Excreted in the Stool from Bile (See further explanation in 3. Results S text below)

## 3. Results

### Model Derivation through Figures and Equations

Based on the above assumptions, Figure 2a–d, and Table 1 and Table 2 in this paper and the following equations in this paper (including in the Appendix A), the synthesis of and demand for 1,25(OH)_2_D_3_ by the many cells/tissues of the body is estimated, and the percent utilization of vitamin D_3_ precursor, 25(OH)D for the synthesis of active vitamin D by the body is determined. Finding a way to estimate the nmols of intracellular 1,25(OH)_2_D_3_ synthesized in the body at any point in time (t) or over a period (t_1_ − t_0_) such as 24 h (in nmols) could be used to identify changing demand for 1,25(OH)_2_D_3_. Determination of the change in total end inactive metabolites of 1,25(OH)_2_D_3_ in the extracellular and intracellular compartments of the body over 24 h (in nmols), plus the number of nmols of 1,25(OH)_2_D_3_ end inactive metabolites excreted over the same 24 h period, provides a way to estimate the 24 h synthesis of intracellular 1,25(OH)_2_D_3_.

Measurement of the nmols of the end inactive metabolites of 1,25(OH)_2_D_3_ excreted from the body in bile/stool during a 24 h period would be difficult. However, an estimate can be made by measuring the end inactive metabolites in a 24 h urine sample and then correcting for the number of inactive metabolites excreted by the stool. Based on previous studies, this model uses the estimation that 25% of 1,25(OH)_2_D_3_ end metabolites are excreted in the urine and 75% are excreted in the stool from bile [41,42,43].

Measuring the nmols of all the intermediate and end metabolites of 1,25(OH)_2_D_3_ in a plasma specimen or in a 24 h urine collection would be tedious. However, measurement of the nmols of the three primary end metabolites of 1,25(OH)_2_D_3_ (i.e., 1-OH-23-COOH-24,25,26,27 Tetranor D_3_ or calcitroic acid, 1,25R(OH)_2_D_3_-26,23S-lactone or calcitriol-26,23-lactone, and 23-COOH-24,25,26,27 Tetranor D_3_ or Calcioic acid) in a plasma specimen and in a 24 h urine may allow the approximate determination of active vitamin D, 1,25(OH)_2_D_3_, synthesis_._ This approximation might be achieved by utilizing the 25% excretion in the urine assumption and the other assumptions listed above (See also the Appendix A).

Since only 25% of the actual excretion of these three vitamin D end metabolites from the body occurs through the urine, the total excretion of end inactive vitamin D metabolites from the body during the 24 h period is four times the excretion in urine. A finding of changing excretion quantities of 1,25(OH)D_3_ end metabolites over 24 h occurring at the same plasma 25(OH)D_3_ concentration in the same individual under different conditions (where the body requires higher or lower amounts of active vitamin D) would support one of the assumptions of this model. Namely, the excretion of fluctuating inactive end metabolites of active vitamin D (and by inference fluctuating active vitamin D synthesis) in the body exists which is partially independent of extracellular plasma active vitamin D precursor 25(OH)D_3_ levels.

The small letter m and its subscripts n (0–10) are used to identify specific metabolites in the vitamin D synthesis pathway. Figure 2a,b describe the enzyme pathways that metabolize 25(OH)D_3_, m_0_, into the production of active vitamin D hormone, 1,25(OH)_2_D_3_, m_1_, and its metabolites. The enzyme pathways that metabolize m_0_ away from the synthesis of active vitamin D into inactive metabolites are part of the body’s regulation of the synthesis of active vitamin D and are also described in Figure 2a,b (see yellow oval).

Figure 2a diagrams some of the intracellular compartment’s enzymatic reactions which result in the accumulation of initial and end metabolites of 25(OH)D_3_ over some arbitrary time interval (t_1_ − t_0_) in nmols. Figure 2b diagrams the rate of synthesis in the intracellular compartment of initial and end metabolites of 25(OH)D_3_, (S_0,n_) in units of nmols/t. Figure 2c presents a simplified diagram of the synthesis, transport of vitamin D metabolites between the intracellular and extracellular compartments, and excretion of vitamin D precursor, active vitamin D, and their end metabolites. Shown with Figure 2c are previously measured extracellular, plasma, and intracellular compartment volumes for an average male and female.

Unlike Figure 1 which describes both the front and backends of the vitamin D metabolic pathways, Figure 2c,d. describe the backend of the vitamin D metabolic pathways. Figure 2d presents a simpler diagram of the synthesis of the end metabolites of 25(OH)D_3_ and 1,25(OH)_2_D_3_ which are produced in some cells/tissues of the body which contain the CYP24A1 enzyme. This enzyme controls the breakdown of active vitamin D, 1,25(OH)D_3_ and the portion of active vitamin D precursor, 25(OH)D_3_, which is diverted away from the synthesis of active vitamin D (See Figure 2a,b). The cells/tissues in diagram, Figure 2d, have arrows pointing out of the cells/tissues to indicate these cell’s ability to metabolize 25(OH)D_3_ and 1,25(OH)_2_D_3_ by the ubiquitous enzyme, CYP24A1. The greater the size of the arrow leaving the cell indicates the greater the concentration of CYP24A1 RNA in the cell, the greater the potential of the cell to synthesize the CYP24A1 enzyme, and ultimately the greater the ability of the cell to metabolize 25(OH)D_3_ and 1,25(OH)_2_D_3_ to their end metabolites.

The pathways from the active vitamin D precursor, 25(OH)D_3_, allow the body to match the synthesis of active vitamin D_3_ to the body’s demand for 1,25(OH)_2_D_3_. If sufficient or more than sufficient vitamin D precursor exists to meet the body’s demand for 1,25(OH)_2_D_3_, then the excess can be diverted away from the synthesis of active vitamin D. In Figure 2d, the red + signs represent the end inactive metabolites of active vitamin D, 1,25(OH)_2_D_3_, or m_1_ and the blue + signs represent the end inactive metabolites of the vitamin D precursor, 25(OH)D_3_, or m_0_ that are diverted away from the synthesis of active vitamin D. The arrows in Figure 2d represent the flow of the end metabolites of 25(OH)D_3_ and 1,25(OH)_2_D_3_ between the intracellular space where these metabolites are synthesized and the extracellular space from which they are secreted into the urine and stool. The flow is assumed to be driven by a concentration gradient (high to low). The size of the arrows represents the quantity of CYP24A1 RNA in the cells/tissues.

Concepts developed in Figure 2a–d. are used to develop formulas in the Appendix A to estimate the amount of active vitamin synthesized by the body through measurement of the total end metabolites of 1,25(OH)_2_D_3_. The equations listed in this text are numbered as they are in the Appendix A where they were derived. The integral of S_0,1_, expressed in Equations (1) and (2) from the Appendix A and repeated below, represents the amount of 1,25(OH)_2_D_3_ produced by cells/tissues of the body over a time interval (t_1_ − t_0_). This integral of S_0,1_ can also be used to estimate the amount of active vitamin D that has been synthesized in the body at any time (t). See Appendix A, Equations (9a), (10a) and (13a). One formula uses the integral of the rate of active vitamin D synthesis, (S_0,1_) over an arbitrary 24 h time interval to estimate the total synthesis of 1,25(OH)_2_D_3_ over 24 h. This synthesis estimate is accomplished by summing the change in the total amount of the end metabolites of 1,25(OH)_2_D_3_ in the extracellular and intracellular compartments and the accumulation of end metabolites of 1,25(OH)_2r_D_3_ in urine and stool over 24 h.

So far, this method has focused on the use of the end metabolites of active vitamin D and the end metabolites of its precursor, 25(OH)D_3_ (which are formed from vitamin D precursor which is diverted away from the synthesis of active vitamin D) in order to evaluate the actual amount of and the need for active vitamin D synthesis. There is one other consideration concerning the synthesis of active vitamin D and the serum concentration of active vitamin D precursor. Gene CYP27B1 of enzyme C1 Hydroxylase is responsible for the synthesis of active vitamin D from its precursor (see Figure 2a and the yellow oval, Figure 2b) and the gene CYP2R1 is responsible for the synthesis of active vitamin D precursor, 25(OH)D_3_, from D_3_. When these genes become mutated and are no longer able to produce functioning 1-α Hydroxylase or 25-hydroxylase enzymes, respectively, severe active vitamin D deficiency can occur [47].

There may be some genetically variable forms of these deficiencies which may impair the production of their hydroxylase enzymes and effect the production of active vitamin D or its precursor without totally stopping the production of these forms of vitamin D. Measurement of the end metabolites of active vitamin D, of its precursor, 25(OH)D_3_, and of the end metabolites of diverted active vitamin D precursor, may help to determine whether either of these two genes and the production rate of their enzymes may be altered.

A second formula has been derived to determine M_1_(t) (Equation (3) below) or the amount of end metabolites of active vitamin D, m_1_, that are present in the body at any point in time (t). The amount of end metabolites of active vitamin D in the body at time (t) is then assumed to represent or to be proportional to the amount of active vitamin D synthesized at that moment. The following numbered equations are taken from derivation of equations in Appendix A that quantify active vitamin D synthesis and demand as well as the percent utilization of the precursor to active vitamin D for active vitamin D synthesis.
(Eq. 1)$$\begin{array}{l} \int_{t_{0}}^{t_{1}}S_{0,1\left(t\right)\;}dt=4.66\mathrm{Vp}[\left(2\beta \right)+1]\times [C_{8,p}\left(t_{0}\;,t_{1}\right)+\;C_{9,p}\left(t_{0}\;,t_{1}\right)+\;C_{10,p}\left(t_{0},t_{1}\right)]+\\ 4[C_{8:10,u}\left(t_{1}\right)\mathrm{Vu}\left(t_{1}\right)] \end{array}$$

for men where V_p_ = 3.0 L, V_u_(t_0_) = 0, and β is initially set at 1.1 (Appendix A)
(Eq. 2)$$\int_{t_{0}}^{t_{1}}S_{0,1\left(t\right)\;}dt=4.4\mathrm{Vp}\;\left[\left(2\beta \right)+1]\times [C_{8:10,\mathrm{cp}}\left(t_{1},t_{0}\;\right)]+4[C_{8:10,u}\left(t_{1}\right)\mathrm{Vu}\left(t_{1}\right)\right]$$

for women where V_p_ = 2.5 L, V_u_(t_0_) = 0, and β is initially set at 1.1 and (Eq. 3) M_1_(t) = M_8:10_(t) = βC_8:10,cp_(t)2V_e_ + C_8:10,cp_(t)V_e_ where V_e_ = 4.66V_p_ for men and 4.4V_p_ for women and β is set at 1.1

In the Appendix A, formulas for a 1,25(OH)_2_D_3_ demand ratio, Drm_1_,_i+e_(t), Equation (4) below and a 1,25(OH)_2_D_3_ 24 h Urine demand ratio, Drm_1_,_u_(t_0_,t_1_), Equation (5) below are also derived. The rationale for using the ratio of the end inactive metabolites of 1,25(OH)_2_D_3_, m_1_, as the numerator and the end inactive metabolites of 25(OH)D_3_, m_0_, as the denominator, is that this ratio will decrease if excess 25(OH)D_3_ exists and is shunted away from the synthesis of active vitamin D when the body’s active vitamin D supply is adequate and vice versa. Knowing this ratio may be helpful clinically to judge whether there is sufficient vitamin D precursor to meet the needs of the body for active vitamin D synthesis. This concept has been introduced previously based on calculating a serum calcitriol/calcifediol ratio using this ratio as an indicator of vitamin D hydroxylation efficiency or as an indicator of relative vitamin D CYP enzyme activities rather than the actual indirect measurement of actual molar substrate 1,25(OH)_2_D_3_/25(OH)D_3_ ratios in the body using the end metabolites of active vitamin D and its immediate precursor [40,48] (see p. 641, Figure 37.3 in Ref. [46]).
(Eq. 4)$${\mathrm{Drm}}_{1,i+e}\left(t\right)=\frac{M_{8:10,i+e}\left(t\right)}{M_{6:7,i+e}\left(t\right)}$$



(Eq. 5)
$${\mathrm{Drm}}_{1,u}\left(t\right)=\frac{M_{8:10,u}\left(t\right)}{M_{6:7,u}\left(t\right)}$$



An increase in m_1_ demand should result in an increase in m_1_ synthesis from m_0_ accompanied by a decrease in diversion of m_0_ away from m_1_ synthesis. (i.e., an increase in the m_1_ demand ratio) This pattern is suggested in the results of a recent study comparing serum vitamin D metabolites between non pregnant women (controls) and pregnant women (at 15 weeks). One group of pregnant women were without the complication of preeclampsia and one group of pregnant women (at 15 weeks) who went on to develop preeclampsia [23]. As stated above, demand for active vitamin D increases during pregnancy. Mean serum active vitamin D levels, m_1_, in these three groups above were, respectively: 85.6, 336.3, and 388.8 pmoles/L. The levels of initial inactive 24,25(OH)_2_D_3_, m_2_, metabolites of m_0_, changed in the opposite direction from active vitamin D synthesis and were, respectively: 9.7, 6.5, and 3.2 nmoles/L These metabolites decreased in parallel with higher m_1_ levels. The differences were not statistically significant but the numbers of subjects were small, respectively: 9, 25, and 25.

A somewhat different pattern was seen in urine samples of one of the initial metabolites of active vitamin D precursor, m_0_ which was diverted away from the synthesis of active vitamin D. Mean metabolite m_2_, 24,25(OH)_2_D_3_, levels in the urine of these three groups were, respectively: 55.4, 84.1, and 35.6 nmols/liter [23]. The mean m_2_ concentrations in the urine samples decreased in the preeclampsia pregnant group compared to the non-preeclampsia pregnant group suggesting higher utilization % in the pre-eclampsia group. The difference between the m_2_ levels between the two pregnant groups was statistically significant. Of note, the other initial intermediate metabolite of m_0_, 23,25(OH)D_3_ or m_3_ which also diverts m_0_ away from the synthesis of active vitamin D was not measurable in the urine samples of this study.

A second method for estimating active vitamin D demand would be determination of the % of vitamin D precursor, 25(OH)D_3_, m_0_, that is used to synthesize active vitamin D, 1,25(OH)_2_ D_3_, m_1_, at one point in time (t) or over 24 h. The first % utilization measurement, Ut_%_m_0_,_i+e_(t), is presented in Equation (6) and represents the percent of m_0_ utilized at a single point in time (t) based on the measurements of the end metabolites, m_8:10_ and m_6:7_ in the intracellular and extracellular compartments. The second measurement of m_0_ percent utilization presented in Equation (7) is based on the measurement of the change in m_8:10_ and m_6:7_ end metabolites in the intracellular and extracellular compartments over 24 h plus the total amount of m_8:10_ and m_6:7_ end metabolites excreted in the same 24 h. (Recall the total body excretion of end metabolites is equal to 4 × the amount of end metabolites excreted in urine. See Appendix A.
(Eq. 6)$$\mathrm{Ut}\%m_{0,i+e}\left(t\right)=\frac{M_{8:10}}{M_{8:10}{+M}_{6:7}}\times 100\%$$



(Eq. 7)
$$\;\int_{t_{1}}^{t_{0}}{\;U^{\prime }t\%m}_{0}\mathrm{dt}=\frac{\int_{t_{0}}^{t_{1}}S_{0,1(t)\;}\mathrm{dt}}{\int_{t_{0}}^{t_{1}}S_{0,1(t)\;}\mathrm{dt}+\int_{t_{0}}^{t_{1}}S_{0,6,7(t)\;}\mathrm{dt}}\times 100\%$$



## 4. Discussion

### 4.1. Measurement of End Metabolites m_8:10_ and m_6:7_

Although the equations above may seem tedious, they will be solved using simple concentration measurements of seven metabolites of vitamin D from plasma and from a 24-h urine sample (including the currently common measurement of plasma m_0_, 25(OH)D_3_ and m_1_, 1,25(OH)_2_D_3_). The seven required metabolite concentration measurements include:**m_0_**—25(OH)D_3_—precursor of active vitamin D—used now to evaluate vit. D status;**m_1_**—1,25(OH)_2_D_3_—active vitamin D (currently available);**m_6_**—25,26,27-Trinorcholecalciferol-24-carboxylic acid—non m_1_ inactive end metabolite of m_0_.**m_7_**—25(OH)D_3_-26-23 Lactone—non m_1_ inactive end metabolite of **m_0_**; **m_8_**—1-OH-23-COOH-24,25,26,27 Tetranor D_3_—inactive end metabolite of m_1_;**m_9_**—1,25R,(OH)_2_D_3_-26-23S—inactive end metabolite of m_1_;**m_10_**—23-COOH-24,25,26,27 Tetranor D_3_—inactive end metabolite of m_1_.Note: See Figure 2a,b.

This molar balance approach depends on the accurate and standardized measurement of the serum end metabolites of active vitamin D and of the diverted end metabolites of its precursor, 25(OH)D_3_. These measurements are the critical foundation of this method. Serum, which is plasma with platelets and clotting factors removed, is used to make these measurements. The serum samples will be formed by spinning down a whole blood sample after clotting has occurred in order to separate the serum from the blood sample cells, platelets, and clotting factors.

Liquid-chromatography-tandem-mass spectrometry (LC-MS-MS) is considered the measurement of choice for vitamin D metabolites [49,50]. With LC-MS-MS, up to twelve vitamin D metabolites have been measured simultaneously [51,52]. Analytical challenges exist with this technology. Sample type, protein precipitation, analyte extraction, derivatization, chromatographic separation ionization, and capabilities of the mass spectrometer must be addressed [50]. Calibration, standardization, and use of internal standards are also important requirements to achieve consistent, accurate results.

Purified standard compounds for only two of the three end metabolites of active vitamin D, m_8_, calcitroic acid, and m_10_, calcioic acid are available commercially [53,54,55]. A purified standard compound for the remaining end metabolite of active vitamin D, m_9_, or calcitriol-26,23-lactone, and standard compounds for the diverted, m_0_ precursor, end metabolites, m_6_ or cholacalcioic acid and m_7_ or 25(OH)D_3_-26-23 lactones will have to be synthesized since these standard compounds are required for the liquid-chromatography-tandem-mass spectrometry measurements of these serum end metabolite measurements.

About 90% of circulating 25(OH)D_3_ is protein bound to vitamin D binding protein (VDBP), albumin, and lipoproteins [49]. The percent binding of active vitamin D end metabolites, and the diverted vitamin D precursor end metabolite, to serum VDBP may not be known. The binding of end metabolites and their relative fat and water solubilities may have to be determined. Measurement of the three end metabolites of active vitamin D and the two end metabolites of the precursor, 25(OH)D_3_ which are diverted away from the synthesis of active vitamin D, may also depend on their actual concentrations in plasma and urine and the range of measurability. The measurements of these end metabolites would be made using the same instruments that are currently measuring other forms of vitamin D.

A source of human CYP24A1 hydroxylase enzyme will be needed to further evaluate and test the proposed assays used by this model. This enzyme will be used to drive the intermediate metabolites to their end metabolites. This reagent (enzyme) will be required to verify that there is not a significant change in total intermediate metabolites during a transitory increase or decrease in active vitamin D synthesis or in the rate of vitamin D precursor diversion. This enzyme is currently available from several commercial sources Biocompare, “https:www.biocompare.com/Search-Biomolecules/?search=CYP24A1 (accessed on 17 July 2020)” [53]. CYP24A1 has been used in previous vitamin D metabolic studies [54].

Vitamin D metabolite measurements have also been made from human spot urine specimens including 25(OH)D_3_ and 24,25(OH)_2_D_3_ [23]. Measurement of human spot urine 1,25(OH)_2_D_3_ and 23,25(OH)_2_D_3_ concentrations were below the limits of detection. The normal volume range of an adult 24 h urine is 800–2000 17mL with a normal intake of 2 L of fluid per day. Since the 24 h urine can be concentrated, concentrating the urine will make the detection of end metabolites in urine easier.

Finally, the technologies used in measuring or monitoring vitamin D metabolites, although challenging, are well established. The only new aspect is adding five different end metabolites of vitamin D to the two currently dominant forms of vitamin D measured, (i.e., active vitamin D_3_, 1,25(OH)_2_D_3_ and its immediate precursor, 25(OH)D_3_).

### 4.2. How Can and Why Should These Vitamin D Parameters Be Used?

Using a molar balance model approach, the following vitamin D parameters can be calculated: (1) an estimate of active vitamin D synthesis in the body at any time (t) and over a 24 h time interval, (2) an active vitamin D demand ratio, and (3) an active vitamin D % utilization. Clinical lab software will perform the calculations listed in Equation (9a) through 18 using the measured concentrations of the end metabolites m_8:10_ and m_6:7_ as well as different constants for men and women. Each of these new vitamin D parameters can then be determined and monitored in many different conditions where the demand by the body for active vitamin D changes dramatically. These parameters cannot be currently determined by simply measuring serum 25(OH)D_3_ and 1,25(OH)_2_D_3_ levels in the serum.

During severe infections caused by many different agents including the SARS-CoV-2 virus, the immune system may require significantly greater synthesis of active vitamin D to suppress the offending agent (i.e., virus) and the inflammatory response caused by the agent [56]. These clinical vitamin D parameters along with other measurements of an individual’s clinical response to their COVID-19 infection will assist clinicians to establish when significant additional vitamin D supplementation is helpful and to determine the required type and dose of vitamin D supplementation.

### 4.3. Molar Balance Model Limitations

The molar balance model depends on several assumptions some of which may need modification as the end metabolites are initially measured and some of the assumptions will not be accurate in all situations. If the extracellular volume or plasma volumes change significantly due to swelling, the assumed volume values will have to be modified. Kidney failure will also prevent the measurement of end metabolites in the urine and may lead to build up of some end metabolites in the body including the extracellular compartment and plasma, and lead to greater excretion of these metabolites in the stool.

## 5. Summary

A new method using a molar balance model is proposed to determine total body active vitamin D, m_1_, levels including synthesis over a 24 h period, the level of demand in an individual for active vitamin D, and the % utilization of vitamin D precursor, m_0_, to synthesize active vitamin D. Determination of these parameters associated with vitamin D levels and synthesis will require the measurement of the end vitamin D metabolites m_6:10_ in both serum and urine using the same equipment and techniques used to measure active vitamin D and its immediate precursor. Knowing this new information may enable clinicians to improve the body’s immune system response to COVID-19 viral infections as well as to other viruses going forward by maximizing the positive effects of active vitamin D on immune system function.

## 6. Conclusions

(1)Because cells/tissues of the body (especially immune system cells) sometimes have dramatically changing requirements for active vitamin D, there may not be a consistent stored amount of vitamin D (as measured by plasma 25(OH)D_3_ levels) or single minimum daily requirement to insure enough active vitamin D synthesis in all circumstances.(2)Knowing the total active vitamin D that has been synthesized at one point in time or that has been synthesized over 24 h, knowing the demand ratio for active vitamin D, and knowing the % utilization of 25(OH)D_3_, m_0_, for the synthesis of active vitamin D may help us to determine changing requirements for active vitamin D at any given time or in different situations the body may encounter.(3)Similarly, knowing these vitamin D parameters may improve our understanding of how-to better dose active vitamin D and active vitamin D precursor supplementation.(4)Suppression of COVID-19 replication or that of another new virus by adequate levels of active vitamin D may not only reduce the effect of the virus on an individual but may also lower the level of contagiousness of an individual with a COVID-19 or other virus infection. Lowering the contagiousness may slow the spread of the virus by lowering the positivity rate within a community [57].

## Figures and Tables

**Figure 1 biomedicines-11-00324-f001:**
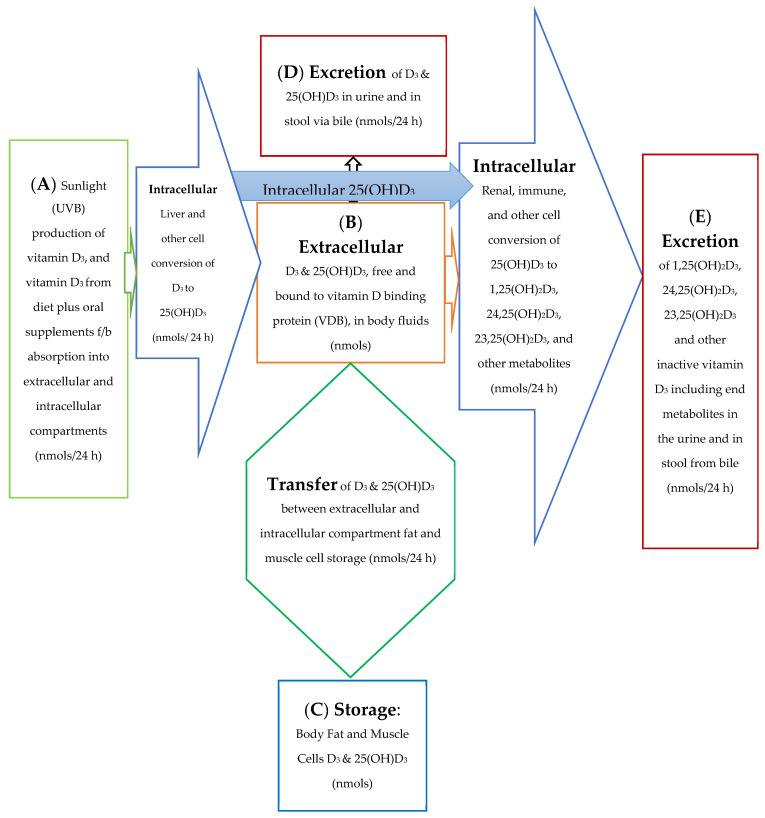
Model for Absorption, Synthesis, Transport, Conversion, Storage, and Excretion of vitamin D Precursors and Metabolites by the Body.

**Figure 2 biomedicines-11-00324-f002:**
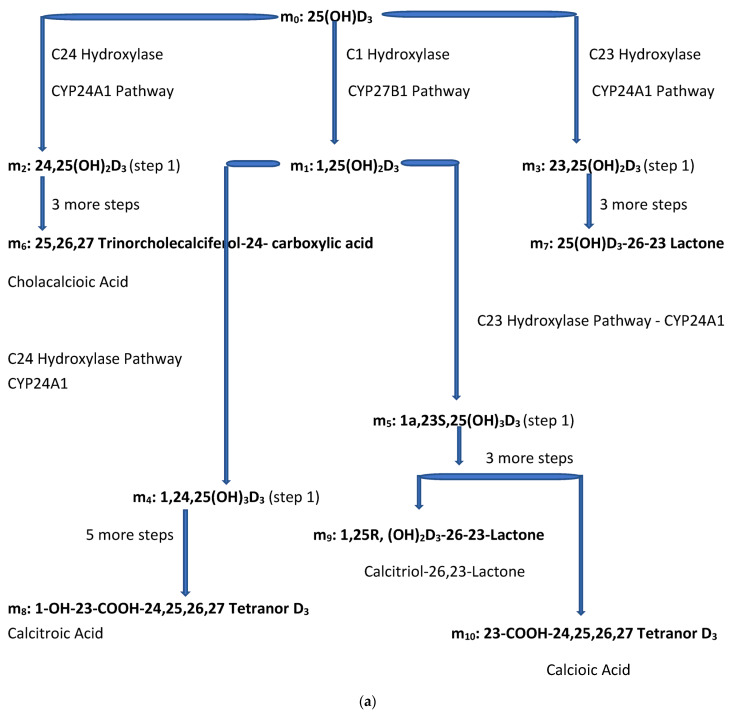
(**a**) Intracellular Vitamin D Pathways Describing the Conversion of 25(OH)D_3_, m_0_, to Its Initial and End Metabolites, m_n_. The C in front of the enzymes represents the carbon atom location which is acted upon by the pathway enzyme. The m_n_ terms describe the initial, some of the intermediate, and the end metabolites of m_0_. The end metabolites (m_6_ to m_10_) pass from the intracellular to the extracellular compartments and are excreted in urine (approx. 25%) and in stool (approx. 75%). (**b**) Depicted in this Flow Diagram is the Intracellular Metabolism of 25(OH)D_3_, m_0_, to Its Initial Metabolites (m_1_, m_2_, and m_3_), and to its End Metabolites in red (m_6_ and m_7_) that are diverted away from active vitamin synthesis. Also depicted is the Intracellular Metabolism of 1,25(OH)_2_, m_1_ and its End Metabolites in red (m_8_, m_9_, and m_10_). In this flow diagram, the m_n_ terms represents the initial and end metabolites of m_0_ (i.e., m_1_ through m_10_). The large bold * symbols represent the enzymes for the synthesis of the initial metabolites of m_0_ (25(OH)D_3_). The small groups of three * symbols represent the enzymes which regulate the amount of end metabolites of active vitamin D hormone, m_1_, and the enzymes which regulate the amount of end metabolites of m_0_ diverted away from m_1_ synthesis in response to a changing need by the body for active vitamin D. (**c**) Model of the Body’s Synthesis, Transport, and Excretion of m_0_ [25(OH)D_3_], m_1_ [1,25(OH)_2_D_3_] and their End Metabolites, m_6_ to m_10_ (nmols/24 h). (**d**) Simple Model of the Body’s Composition of Non m_1_ End Metabolites from [25(OH)D_3_], m_6_ and m_7_ (*), and End Metabolites from m_1_ [1,25(OH)_2_D_3_], m_8_ to m_10_ (+), at any point in time, t, in moles. The open space in the box is the extracellular space and the circles/ovals represent the intracellular space/tissues (IC/T). The size of the arrows indicates the amount of CYP24A1 enzyme RNA in the IC/T and the potential for CYP24A1 synthesis and ultimately the ability for metabolism of 25(OH)D_3_ and 1,25(OH)_2_D_3_ to their end metabolites.

**Table 1 biomedicines-11-00324-t001:** Intracellular and Extracellular Compartment Volume Analysis.

Water Makes up Approximately 60% of Body Weight in the Male and 55% Body Weight in the Female
For a 55 kg female:
Extracellular Compartment Volume (**V_e_**) = ’s about 11 L [including plasma volume (**V_pla_**) 2.5 L, + Interstitial volume (**V_inter_**) 8.5 L + transcellular volume (**V_trans_**) 0.4 L–aver.]
2. Intracellular Compartment Volume (**V_i_**) =’s about 22 L
3. **V_i_** = ’s about 2**V_e_** and in females **V_e_** equals about 4.4V_pla_ Ref. [45]
4. Approx. mean HCT 40% (range 36–44%) my.clevelandclinic.org/health/diagnostic17683-hematocrit/results-and-follow-up, assessed on 8 March 2020
For a 70 kg Male:
5. Extracellular Compartment Volume (**V_e_**) = ’s about 14 L [including plasma volume (**V_pla_**) 3.0 L, + Interstitial volume (**V_inter_**) 10.5 L + transcellular volume (**V_trans_**) 0.5 L–aver.]
6. Intracellular Compartment Volume (**V_i_**) = ’s about 28 L
7. **V_i_** = ’s about 2**V_e_** and in males V_e_ equals about 4.66V_pla_ Ref. [46]
8. Approx. mean hematocrit, HCT, 45.5% (range 41–50%) my.clevelandclinic.org/health/diagnostic17683-hematocrit/results-and-follow-up assessed on 8 March 2020

Differences between average male and female volumes are as follows: Plasma volume—males have 20% greater volume based on average size; Extracellular volume—males have 27% greater volume based on average size; Intracellular volume—males have 27% greater volume based on average size; Because of the differences in mean % hematocrit as well as physical size between men and women, the total % difference in aqueous volume (extracellular + intracellular) between men and women would be a difference of 42 L/33 L × 100% or 127%.

**Table 2 biomedicines-11-00324-t002:** Total Male and Female Intracellular, Extracellular, and Plasma Volumes.

For Males V^e+i^ = 42 L	For Females V^e+i^ = 33
V_i_ = 28 L	V_i_ = 22 L
V_e_ = 14 L	V_e_ = 11 L
V_p_ = 3 L	V_p_ = 2.5 L

## Data Availability

Not Applicable.

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
