# Peer review of "A Mathematical Model for Determining the Body’s Fluctuating Need for and Synthesis of Active Vitamin D"

_biomedicines, 2023, doi:10.3390/biomedicines11020324_

Round 1

Reviewer 1 Report

The present paper aimed to demonstrate that serum active vitamin D levels often do not correlate with vitamin D precursor levels and may not correlate well with active vitamin D synthesis in the body.

A few changes are needed, as follows:

Please explain every abbreviation before using it, including figure legends.

Please provide some final conclusions for your manuscript!

Reviewer 2 Report

The manuscript titled “Determination of 1,25(OH)2D3 Synthesis and Demand, and Percent 25(OH)D3 Utilization” attempted to demonstrate a mathematical model involving active vitamin D and precursor to it.

 As such the work in this manuscript is interesting. However, there are some deficiencies in the manuscript which need to be addressed to make it publishable:

 1.      The title of the manuscript is bit confusing in the current form and can be modified to include the model.

2.      Introduction- authors highlighted the deficiency of vitamin D in COVID-19 as a prime point in the introduction. This is important that authors summarize the multitude roles of vitamin D in different conditions.

3.      Quality of the figures should be improved. The fonts and schematics are sometimes unclear.

4.      There are shorter paragraphs without developing a flow concepts. They need to be combined with concept similarity. 

5.      Potential application of this model and calculations should be highlighted in the discussion section.
